# The Effect of COVID-19 Outbreak and Incidence on the Health-Related Behaviors and Depression of Gyeongnam Residents in Republic of Korea

**DOI:** 10.3390/medicina59091672

**Published:** 2023-09-16

**Authors:** Young-Mi Seo, Ki-Soo Park

**Affiliations:** 1Department of Health Management, Graduate School of Public Health, Gyeongsang National University, Jinju 52727, Republic of Korea; sechki486v@naver.com; 2Department of Preventive Medicine, Institute of Medical Science, College of Medicine, Gyeongsang National University, Jinju 52727, Republic of Korea

**Keywords:** COVID-19, social distancing, health behavior, depression

## Abstract

*Background and Objectives*: This study aimed to investigate whether the occurrence of COVID-19 brought about changes in the health behaviors and depression levels of residents in Gyeongnam in Republic of Korea, and whether the prevalence of COVID-19 was related to differences in health behaviors and depression levels among different regions. *Materials and Methods:* The researchers utilized raw data from the 2019–2020 Community Health Survey in Gyeongnam and conducted analyses using SPSS 25.0. The study included a total of 35,880 participants from 18 cities and counties in the Gyeongnam region (17,942 participants in 2019 and 17,938 participants in 2020). *Results*: The results of the comparative analysis between pre- and post-COVID-19 occurrence showed that, after the occurrence of COVID-19, the smoking cessation rate and monthly alcohol consumption rate among current smokers decreased, while the high-risk drinking rate increased. The rate of physical activity (walking) increased, but the prevalence of depression experiences and depressive symptoms also increased. Furthermore, the comparative analysis between areas with a higher number of COVID-19 cases and those with a lower number of cases revealed that areas with a higher number of cases had higher monthly alcohol consumption rates, as well as a higher prevalence of depression experiences and depressive symptoms. *Conclusions*: Considering that the occurrence and severity of COVID-19 had significant impacts on the health behaviors and depression levels of residents in Gyeongnam, this highlights the need for active intervention and management by the national and local governments in response to the occurrence and management of infectious diseases, including COVID-19, to address the health status and health behaviors of the local population.

## 1. Introduction

COVID-19 is an epidemic respiratory disease caused by a new type of coronavirus (severe acute respiratory syndrome coronavirus 2, SARS-CoV-2) [1]. In Republic of Korea, the first case of coronavirus was reported on 20 January 2020. Since then, the situation has gradually worsened, leading to an escalation of the infectious disease crisis alert level from ‘caution’ to ‘warning’ and, eventually, to ‘serious’ [2]. In response to the spread of COVID-19, a high-level ‘social distancing’ campaign was implemented [3]. The population was encouraged to stay at home and limit social interactions as much as possible. Various measures, such as restricting business hours and the number of people allowed to gather, were implemented at different levels [4]. These measures had become part of everyday life, had brought about substantial societal changes, and had affected individuals’ mental health in a far-reaching manner.

Furthermore, the isolation brought about by social distancing could lead to a decrease in outdoor activities and exercise and be accompanied by excessive smoking and alcohol consumption. The elderly were particularly adversely affected by social distancing policies as village halls and welfare centers, places for the elderly to socialize and exercise, were closed. Additionally, health care for the elderly was curbed, and their physical activity levels and nutrition management became issues of concerns.

A number of cross-sectional studies have investigated different aspects of health-related behaviors and mental health during the pandemic, including effects on physical activity [5,6] and mental health. [7]. Although there were few longitudinal studies, published studies suggested that certain aspects of mental health, such as psychological distress, increased significantly during the early stages of the COVID-19 pandemic compared to pre-pandemic levels [8,9]. Some changes in health-related behaviors and mental health may be temporary and may return to pre-pandemic levels once COVID-19 public health restrictions begin to ease, but some of these changes are likely to be more permanent. This can negatively affect a person’s long-term health. That being said, it is important to highlight which health-related behaviors and aspects of mental health are being impacted through the continuation of COVID-19 public health restrictions.

As of 31 August 2020, after the implementation of community health surveys, differences in COVID-19 incidence across Gyeongnam’s regions could be identified. The regional COVID-19 incidence differences have resulted in the application of varied social distancing guidelines and quarantine rules. At this time, the incidence of COVID-19 in Republic of Korea was about 0.5 per 100,000 people. We hypothesized that, due to the occurrence of COVID-19, health behavior practices would vary across different regions. We also postulated that public mental health, as measured by incidence and prevalence of depression and depressive symptoms, would similarly vary.

Although there are many investigations evaluating the health behaviours and mental health of individuals during the COVID-19 pandemic, there is a need of further knowledge in the Korean context using large-scale national data collected during the COVID-19 pandemic. For example, there are knowledge gaps concerning regional occurrences of infectious diseases during the COVID-19 epidemic, and similar gaps are present in our understanding of the effects of COVID-19 on health-related behaviors and public mental health. We endeavored to fill this gap for health-related behaviors, smoking, alcohol consumption, and participation in physical activity, and public mental health.

In this context, the research questions of this study are the following:We examined the impact of COVID-19 on the practice of health-related behaviors and depressive symptoms by comparing the changes in the prevalence of these practices and symptoms. This examination involved a comparison of health survey data before (2019) and after (2020) the COVID-19 outbreak.We evaluated the difference in health behaviors and depressive symptoms between low-risk and risky COVID-19 regions in 2020.

## 2. Materials and Methods

### 2.1. Study Design and Population

This study is a cross-sectional study analyzing secondary data collected from the country. We used Gyeongnam province data from the 2019–2020 Community Health Survey (CHS). This survey is a nationwide sample survey conducted by the Republic of Korea Centers for Disease Control and Prevention in which a representative sample is selected by city, county, and district every year. The survey is conducted annually from 16 August to 31 October for household members aged 19 or older living in the selected sample households [10]. A trained researcher conducts a 1:1 interview (computer-assisted personal interview, CAPI) with a laptop computer, investigating the health-related behavioral practices and symptoms of depression among residents [10]. Stratification variables, clustering variables, and individual weights are applied when analyzing data with a composite sample design survey in which a part of the population that can reflect the entire population is extracted.

The subjects used in the analysis were 17,942 in 2019 (male = 44.2%, female = 55.8%) and 17,938 in 2020 (male 44.9%, female 55.1%), residents of 18 cities and counties in the Gyeongnam region.

The study was conducted according to its purpose after the review and approval of the Institutional Review Board of Gyeongsang National University (IRB No. GIRB-G21-X-0058).

### 2.2. Materials

#### 2.2.1. Dependent Variables

Smoking, alcohol consumption, physical activity, and depression prevalence were dependent factors.

The current smoking rate was calculated as the percentage of current smokers among those who had smoked five packs or more in their lifetime, and the male current smoking rates were calculated as the proportion of current smokers among those who had smoked five or more cigarettes during their lifetime. The rate of current smokers’ cessation attempts was calculated as the percentage of current smokers who had attempted to quit smoking for more than 24 h in the past year, regardless of the type of cigarette used.

The monthly alcohol consumption rate was calculated as the percentage of survey subjects who imbibed at least once a month during the previous year. The high-risk drinking rate was calculated as the percentage of respondents who answered “yes” to the high consumption question. For men, this question defined high consumption as seven or more drinks twice a week or more during the previous year. For women, this quantity was defined as five or more drinks over the same frequency.

The rate of moderate or high-intensity physical activity is the number of people who have engaged in vigorous physical activity (20 min or more a day, 3 days or more in the past week) or moderate physical activity (30 min or more a day, 5 days or more in the past week); this was calculated as a fraction of one person. The level of physical activity was assessed using the Korean version of the International Physical Activity Questionnaire (IPAQ) [11,12]. Participants were asked to respond to questions indicating performance of high- or moderate-intensity exercise ten minutes or more in the past week. These questions also included individuals’ walking frequency. High-intensity physical activity refers to activities such as running, mountain climbing, and brisk cycling, while moderate-intensity physical activity refers to activities such as slow swimming or participation in doubles tennis or badminton. Walking activity time refers to all walking activities, including commuting to and from work, going to and from school, moving, and exercising. The walking activity rate was calculated as the percentage of survey subjects who practiced walking for at least 30 min a day for at least 5 days a week during the most recent week.

The proportion of individuals who experienced depressive symptoms, such as sadness or despair, severe enough to significantly interfere with daily life for a continuous period of two weeks or more in the past year was calculated as the proportion of experiencing depressive feelings. The prevalence of depression symptoms was calculated using the Korean version of the Patient Health Questionnaire-9 (PHQ-9), a screening tool for depression. The proportion of individuals scoring 10 or higher on the PHQ-9 was used in analysis. This questionnaire consists of nine questions and a four-point Likert scale corresponding to the individual’s status in the past two weeks. This scale ranged from 0 (not at all) to 3 points (nearly every day). The highest possible score is 27 points, and a score of 10 or more indicates the presence of a depressive disorder [13,14].

#### 2.2.2. Covariables

The independent variables used in the analysis were categorized based on the distinction between pre-COVID-19 (2019) and COVID-19 epidemic period (2020) and the regional severity of the COVID-19 outbreak in the Gyeongnam province.

Adjustment variables include gender, age, education level, monthly income, and marital status. Age was categorized into two groups, ‘less than 65’ and ‘65 or older’. Education level was categorized into four groups, ‘below elementary school graduation’, ‘middle school graduation’, ‘high school graduation’, and ‘university graduation or higher’, for analysis. Marital status was analyzed by being classified into two categories: ‘with a spouse’ and ‘without a spouse (e.g., divorce or bereavement)’. Monthly income level was classified into four categories, ‘less than 1 million won’, ‘1 million to 2.99 million won’, ‘3 million to 4.99 million won’, and ‘more than 5 million won’, for analysis. 

### 2.3. Data Analysis

The collected data were analyzed by a complex sample design method reflecting stratification, colony extraction, and individual weights using the SPSS version 25.0 statistical program.

General characteristics of Gyeongsangnam-do residents were analyzed by descriptive statistics such as real numbers, percentages, averages, and standard deviations, and differences in health behavior and depression prevalence by year were analyzed by *t*-test and chi-square test.

In order to examine the correlation between the occurrence of COVID-19 and its impact on the mental health status of residents in the Gyeongsangnam-do region, a multivariable logistic regression analysis was conducted and all entered variables were considered. Dependent variables included health behavior and depression prevalence, and multivariate logistic analysis was performed for each dependent variable; and modifier variables included gender, age, education level, monthly income level, and marital status. The model fit was determined using the Hosmer and Lemeshow method.

To classify regions based on the difference in the number of confirmed COVID-19 cases, the number of confirmed COVID-19 cases per city or county was used as a criterion. The criteria for classification were based on the data released by the Korea Centers for Disease Control and Prevention’s Central Disease Control Headquarters in August 2020. At that time, there were approximately 300 daily cases nationwide; therefore, the social distancing level was raised from Level 2 to Level 3 for enhanced management. “Risky” areas were determined to be those urban areas with an accumulated occurrence of ten or more cases, or rural areas with five or more cases. Therefore, this study also used this classification criterion. Accordingly, multivariable logistic regression analysis was conducted to determine presence of differences in the health-related behaviors and depression prevalence of residents among regions with variable COVID-19 prevalence. Gender, age, education level, monthly income level, and marital status were included as correction variables at this stage of the analysis. Multivariate logistic analysis was performed for each dependent variable. The model fit was determined using the Hosmer and Lemeshow method. Statistical significance level was set at *p* < 0.05.

## 3. Results

### 3.1. General Characteristics

Table 1 shows the general characteristics of residents in Gyeongnam in 2019 and 2020.

There was no statistical difference in the sociodemographic characteristics of the subjects in 2019 and 2020. The average age of the 17,942 respondents in 2019 was 57.2 ± 16.96 years old; women accounted for 55.8% of the survey population. By education level, high school graduates were the most represented, accounting for 32.0% of the respondents; as for marital status, having a spouse accounted for 67.7% of those surveyed, and those with a household income of 1 million won or more and less than 3 million won accounted for the most, 32.5%. And, for the 2020 survey, the average age of the 17,938 participants was 56.8 ± 17.32 years old, where 55.1% were women. High school graduates comprised the highest percentage among the groups, 33.3%; as for marital status, having a spouse accounted for 64.1% of respondents, and household income of 1 million won or more and less than 3 million won was the highest at 32.8%.

Among health behaviors, the current smoking proportion and the male current smoking proportion were not statistically significant. However, the current smokers’ smoking cessation attempt proportion was significant at 52.2%, a decrease of 23.1% compared to 2019 (*p* < 0.001). Compared to 57.2% in 2019, the monthly drinking proportion in 2020 was 52.5%, which was significantly different (*p* < 0.001), and the high-risk drinking proportion was 17.1%, an increase of 2.6% compared to 14.5% in 2019 (*p* < 0.001). In physical activity, there was no significant change in the rate of moderate or higher physical activity, but the rate of walking practice was significantly different at 42.2% compared to 35.9% in 2019 (*p* < 0.001).

The depression symptom experience proportion was 6.3%, a significant increase of 1.4% compared to 4.9% in 2019 (*p* < 0.001), and the prevalence of depression (PHQ-9 score ≥10) was statistically significant at 15.0%, an increase of 2.3% compared to 12.7% in 2019 (*p* = 0.001).

### 3.2. Changes in the Health Behaviors and Depression among Residents before and after the Occurrence of COVID-19

Figure 1 shows the results of multivariable logistic regression analysis on changes in the health behavior and depression prevalence of residents of Gyeongnam before (2019) and after (2020) the outbreak of COVID-19.

Current smoking rates and male current smoking rates did not show significant results between the two groups before (2019) and after (2020) the outbreak of COVID-19 (*p* = 0.769 and *p* = 0.572, respectively). The odds ratio of attempts to cease smoking was 0.395, which was significantly lower in 2020 than in 2019 (*p* < 0.001).

The monthly alcohol consumption rate, with an odds ratio of 0.813, decreased after COVID-19 occurred (2020) compared to before (2019) (*p* < 0.001). However, the odds ratio for high-risk drinking was 1.236, significantly higher after the COVID-19 outbreak (*p* < 0.001).

There was no significant difference in moderate- to high-intensity physical activity participation before and after COVID-19 onset (*p* = 0.910). Walking activity increased after the outbreak, with an odds ratio = 1.504 (*p* < 0.001). 

The depression experience rate odds ratio was 1.515, and the rate of experiencing depression increased after the outbreak of COVID-19 (*p* < 0.001). The prevalence of depressive symptoms also increased, with an odds ratio of 1.272, after the onset of COVID-19 (*p* = 0.001).

### 3.3. Differences in Health Status and Depression among Regions Based on the Difference in the Number of Confirmed COVID-19 Cases

Table 2 shows the comparison results between “risky” and “low-risk” COVID-19 regions.

Among health behaviors, there was no significant difference between the current smoking proportion and male current smoking proportion according to the difference in the confirmed COVID-19 cases (*p* = 0.125, *p* = 0.316). The proportion of attempts to quit smoking was significantly higher in areas with a large number of confirmed COVID-19 cases by 5.0% (*p* = 0.022).

The monthly drinking proportion was significantly higher in risky areas by 9.4% (*p* < 0.001), and the high-risk drinking proportion was also significantly higher by 3.6% (*p* < 0.001) in risky areas.

In physical activity, there was no significant change according to the difference in the confirmed COVID-19 cases in the proportion of moderate or higher physical activity and walking activity (*p* = 0.385, *p* = 0.382).

Regarding depression, the experience proportion of depressive symptoms was significantly higher in the risky areas by 1.9% than that in low risk areas (*p* < 0.001), and the prevalence of depression was significantly different by 5.1% (*p* < 0.001).

Figure 2 shows the multivariable logistic analysis comparison results between “risky” and “low-risk” COVID-19 regions.

There was no significant difference in the current smoking, male current smoking, and smoking cessation attempt proportions according to the difference in the number of confirmed COVID-19 cases (*p* = 0.212, *p* = 0.402, and *p* = 0.101, respectively).

The monthly drinking rate odds ratio was 1.111, and higher in “risky” than “low-risk” COVID-19 areas (*p* = 0.017). However, there was no significant difference in the high-risk drinking rate (*p* = 0.643).

There was no significant difference in moderate or higher physical activity participation and walking frequency according to the difference in the number of confirmed cases of COVID-19 (*p* = 0.726 and *p* = 0.570, respectively).

The odds ratio for experiencing depression was 1.614. Participants in “risky” COVID-19 areas were more likely to experience depression (*p* < 0.001). The prevalence of depressive symptoms also showed an odds ratio of 1.688, which was higher in areas with more confirmed cases of COVID-19 than in areas without (*p* < 0.001).

## 4. Discussion

This study was conducted to determine the effects of COVID-19 on survey subjects’ symptoms of depression and on their health-related behaviors. We also aimed to determine these effects across “risky” and “low-risk” geographical regions.

The comparative analysis before and after the occurrence of COVID-19 revealed that the occurrence of COVID-19 was associated with a decrease in the smoking cessation attempts among current smokers. This suggests that the isolation and increased time spent at home due to social distancing may increase stress and induce depression. A natural consequence of the increased stress is increased smoking frequency [15]. The negative effect of the outbreak on smoking cessation may have been exacerbated by the 48% reduction in services of health institutions that operate smoking cessation clinics [16]. Smoking is considered a major risk factor for individuals with underlying health conditions, and toxic substances like nicotine in cigarettes can impair lung immune cells and weaken the immune system, thereby exacerbating symptoms of COVID-19. Therefore, the decrease in the smoking cessation attempt rate may reflect a negative impact on COVID-19 prevention as continued smoking may contribute to the worsening of symptoms caused by the virus [17].

The monthly alcohol consumption rate was lower than before the outbreak of COVID-19, but the rate of high-risk drinking increased after the outbreak of COVID-19. This suggests that the reduction of dinners and gatherings due to regulations on telecommuting and private gatherings due to social distancing [18] and restrictions on business hours may have affected the decrease in monthly drinking rates. However, the high-risk alcohol consumption rate has increased after the occurrence of COVID-19. According to the survey results conducted by the Ministry of Food and Drug Safety in 2020 on the alcohol consumption habits of our citizens, the consumption of alcohol alone, “hon-sul”, and drinking at home, “hom-sul”, increased in response to the impact of COVID-19, indicating the emergence of a new type of drinking culture [19]. In addition, personal factors caused by stress and anxiety affected problem drinking [20]. For proper health management, having social drinking habits is important. Since smoking and excessive drinking can have negative effects on COVID-19 prevention, the government needs to maintain policies that support moderation, including smoking cessation, to promote overall well-being [21].

The walking practice rate increased from before the outbreak of COVID-19. This may indicate that individuals prefer easy-access walking, which can be performed alone despite the closure of group sports facilities and restrictions on outdoor activities [22]. Furthermore, in response to the hiatus in community-based health programs such as exercise classes and music classes, individual participation in outdoor walking demonstrates a desire to maintain a healthy lifestyle [23]. The prevalence of walking and light outdoor exercise also increased because it was a way to deal with boredom and other unwanted psychological symptoms. Governments should be careful about imposing restrictions (e.g., closing recreational facilities and playgrounds) that prevent individuals from engaging in desirable physical activity.

The rate of experiencing depression and the prevalence of depressive symptoms increased after COVID-19 onset. This could be due to various risk factors, such as mental stress and anxiety caused by COVID-19, affecting residents’ feelings of depression [24]. As the epidemic prolongs, there is an increase in depression progression and suicide risk. Therefore, there is a need for central and local governments to establish and implement health policies to prevent depression progression and to install “safety nets” through focused efforts [25].

“Risky” COVID-19 geographical areas had a higher monthly alcohol consumption rate than “low-risk” ones. The depression experience rate and depression prevalence were also increased. This suggests regional infectious disease severity may have a negative impact on alcohol consumption and feelings of depression. We have inferred that various restrictive situations caused by the COVID-19 pandemic increase the risk of depression and, as a result, may lead to higher dependence on alcohol [26].

### Limitations

This study has several limitations. First, it was impossible to definitively explain whether health behaviors and depression were affected by COVID-19. In other words, the health-related behaviors and depression of Gyeongnam residents might decrease or increase due to other factors. In order to solve these limitations, it will be necessary to study whether health behaviors and depression change after COVID-19 decreases.

Second, adjusting for other confounding factors that may influence the changes in health behaviors and depression among residents in the Gyeongnam region would have been ideal. However, due to the limitations of the data, this was impossible. More detailed research that considers additional relevant variables is needed.

Third, since we only investigated data from Gyeongnam province, this study’s findings could be different in other provinces.

Despite these limitations, the significance of this study lies in comparing the impact on the health behaviors and depression of residents based on differences in the COVID-19 outbreak. During epidemics of infectious diseases, it is necessary not only to control infectious diseases, but also to manage the mental health status and health-related behaviors of residents, based on the degree of infectious disease occurrence.

The results of this study suggest the need for national and local government intervention in promoting a healthy lifestyle and managing mental health to mitigate the negative impact and promote health behaviors among the population affected by COVID-19. Furthermore, the study emphasizes the importance of efforts to reduce disparities in health behaviors and depression among residents, which may arise from differences in the prevalence of COVID-19 across regions. Additionally, just as social distancing campaigns were implemented, government intervention in promoting healthy lifestyles and managing mental health needs to be tailored to the situation.

## 5. Conclusions

The occurrence of COVID-19 has had a significant impact on the health behaviors and depression levels of residents in Gyeongnam. Therefore, active intervention and management by the national and local governments are necessary concerning the occurrence and management of infectious diseases, including COVID-19, to address the health status and health behaviors of the local population. In other words, the central government should create and operate clear guidelines for controlling infectious diseases, and local governments should quickly implement infectious disease control according to the central guidelines, and provide customized health care. For example, smoking cessation, drinking temperance programs, and the activation of mental health management using digital technology, and distribution of various physical activities that can be enjoyed alone should be considered in the event of an infectious disease outbreak.

## Figures and Tables

**Figure 1 medicina-59-01672-f001:**
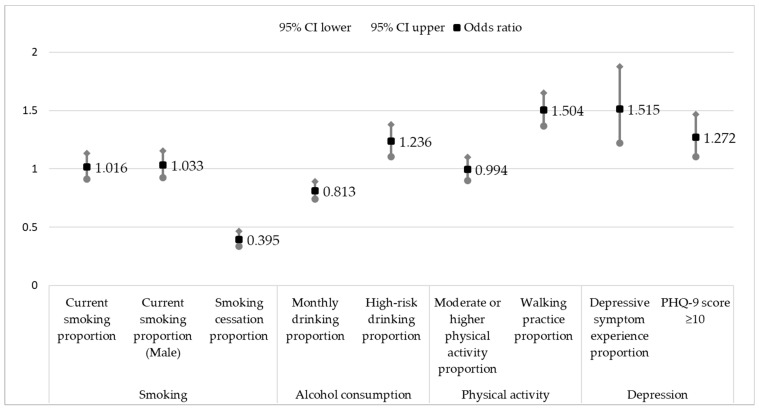
Multivariable logistic regression analysis of changes in the health behaviors and depression among residents before and after the occurrence of COVID-19. (N = 35,880). Adjusted for age, gender, education level, marital status, and income; CI = confidence interval.

**Figure 2 medicina-59-01672-f002:**
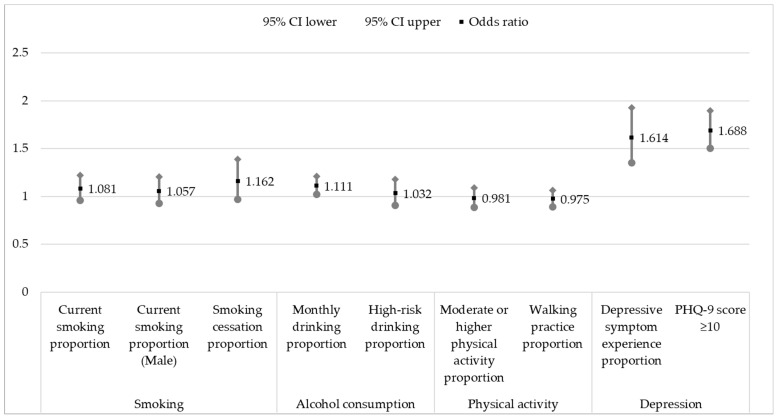
Multivariable logistic regression results for differences in health behaviors and depression according to regional differences in COVID-19 cases. (N = 17,938). Reference = COVID-19 low-risk area; adjusted for age, gender, education level, marital status, and income; CI = confidence interval.

**Table 1 medicina-59-01672-t001:** General characteristics of Gyeongnam residents in 2019–2020.

Variable	2019 N (%)	2020 N (%)	*p* Value
Total	17,942	17,938	
Age (years)	57.2 ± 16.96	56.8 ± 17.32	0.751
≥65	6582 (36.7)	6499 (36.2)	0.854
<65	11,360 (63.3)	11,439 (63.8)	
Gender			0.865
Male	7937 (44.2)	8051 (44.9)	
Female	10,005 (55.8)	9887 (55.1)	
Education			0.342
≤Elementary School	5467 (30.5)	5173 (28.9)	
Middle school	2256 (12.6)	1985 (11.1)	
High school	5737 (32.0)	5977 (33.3)	
≥University	4478 (25.0)	4788 (26.7)	
Spouse			0.124
Yes	12,144 (67.7)	11,501 (64.1)	
No	5792 (32.3)	6432 (35.9)	
Monthly household income (10,000 won)			0.846
<100	3801 (21.2)	3894 (21.7)	
100–299	5831 (32.5)	5875 (32.8)	
300–499	4314 (24.0)	3877 (21.6)	
≥500	3885 (21.7)	4111 (22.9)	
Smoking			
Current smoking proportion	2806 (18.2)	2791 (18.2)	0.978
Current smoking proportion (Male)	2545 (33.8)	2566 (33.9)	0.867
Smoking cessation proportion	5054 (75.3)	1462 (52.2)	<0.001
Alcohol consumption			
Monthly drinking proportion	8412 (57.2)	7788 (52.5)	<0.001
High-risk drinking proportion	2050 (14.5)	2116 (17.1)	<0.001
Physical activity			
Moderate or higher physical activity proportion	4705 (26.1)	4490 (25.7)	0.605
Walking practice proportion	4490 (25.7)	7149 (42.2)	<0.001
Depression			
Depressive symptom experience proportion	6080 (35.9)	936 (6.3)	<0.001
PHQ-9 score ≥ 10	7149 (42.2)	2275 (15.0)	0.001

**Table 2 medicina-59-01672-t002:** Difference of health behaviors and depression of residents according to the difference in COVID-19 confirmed cases in 2020. (N = 17,938).

Variable		Low-Risk Regions	Risky Regions	Difference	*p*-Value
Smoking	Current smoking proportion	17.3	18.4	1.1	0.125
	Current smoking proportion (Male)	32.8	34.2	1.4	0.316
	Smoking cessation proportion	48.0	53.0	5.0	0.022
Alcohol consumption	Monthly drinking proportion	44.7	54.1	9.4	<0.001
	High-risk drinking proportion	14.1	17.7	3.6	<0.001
Physical activity	Moderate or higher physical activity proportion	25.0	25.8	0.8	0.385
	Walking practice proportion	42.9	42.0	-0.9	0.382
Depression	Depressive symptom experience proportion	4.7	6.6	1.9	<0.001
	PHQ-9 score ≥ 10	10.8	15.9	5.1	<0.001

Chi-square test.

## Data Availability

The data are contained within the website (https://chs.kdca.go.kr/chs/main.do) (Accessed on 10 November 2021).

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
