# Peer review of "The Effect of COVID-19 Outbreak and Incidence on the Health-Related Behaviors and Depression of Gyeongnam Residents in Republic of Korea"

_medicina, 2023, doi:10.3390/medicina59091672_

Round 1
Reviewer 1 Report
The manuscript of Seo and Park is well-written. The authors provide data on changes in the behavior of people in South Korea during the COVID-19 pandemic. There are a few comments:
1. To improve the readability, I would suggest presenting some of the data in graphs instead of tables.
2. Authors should discuss how mass COVID-19 vaccination influenced depression, smoking, and drinking habits. There are a few interesting articles suggesting that new vaccines have caused anxiety and an increase in alcohol consumption (10.1016/j.janxdis.2022.102598; 10.18549/PharmPract.2022.3.2689; 10.3390/pathogens12020163; doi.org/10.1016/j.eclinm.2022.101375).
3. Despite the obvious for many readers location, the authors should change "Korea" to "South Korea" in the title and text since data was collected in South Korea only and it is an international name of the country.
Author Response
- To improve the readability, I would suggest presenting some of the data in graphs instead of tables.
-> Tables 2 and 4 have been modified to Figures 1 and 2.
- Authors should discuss how mass COVID-19 vaccination influenced depression, smoking, and drinking habits. There are a few interesting articles suggesting that new vaccines have caused anxiety and an increase in alcohol consumption (10.1016/j.janxdis.2022.102598; 10.18549/PharmPract.2022.3.2689; 10.3390/pathogens12020163; doi.org/10.1016/j.eclinm.2022.101375).
-> This study did not include the part you mentioned because it was data before vaccination began in Korea.
- Despite the obvious for many readers location, the authors should change "Korea" to "South Korea" in the title and text since data was collected in South Korea only and it is an international name of the country.
-> Corrected
Reviewer 2 Report
The researchers worked hard to develop a study focused Effect of COVID-19 Outbreak and Incidence on the 2 Health-related Behaviors and Depression of Gyeongnam. This is important work and adds to the literature.
Introduction & Literature Review
· The Introduction section covers the background of the study. The rationale of the study is also explained clearly in the study.
· In the literature review chapter, previous research is well supported. The literature review is consistent, all-inclusive and critical arguments are well written.
Method
· Methodology is also described well in the study.
Results & Discussion
• The discussion section lacks a thorough discussion of the findings. The authors did a nice job of putting results in context but needs to address more how their research extends the literature. Add a paragraph or two that addresses what meaning is made from the findings cumulatively.
Conclusion
Conclusion is concise and well written.
Author Response
The discussion section lacks a thorough discussion of the findings. The authors did a nice job of putting results in context but needs to address more how their research extends the literature. Add a paragraph or two that addresses what meaning is made from the findings cumulatively.
- Corrected
Reviewer 3 Report
This is a great paper that covers a broad spectrum of health behaviours and psychological outcomes using a large dataset from Korea during the COVID-19 pandemic. I thought the paper, specifically the introduction, results, and limitations sections were nicely presented. I did feel the abstract, conclusion, model building/fitting, and some portions of the discussion should be revised. Please see below for my specific comments.
1. Introduction. "Data on health behaviors and mental health states during the COVID-19 epidemic are abundant, but research making maximal use of these data is lacking" As a researcher in both of these areas, I can confidently state this is not true. Instead, you could change the narrative to focus on South Korea. For example, say something along the lines of "Although there are many investigations evaluating health behaviours and mental health of individuals during the COVID-19 pandemic, there is a need to further knowledge in the Korean context using large-scale national data collected during the COVID-19 pandemic." because you are contributing useful evidence.
2. Methods. Thank you for describing your analysis in some detail. A few questions to clarify: (a) By "multiple logistic regression", do you mean multivariable logistic regression or several different univariable models? (b) What model building approach did you take? (c) Did you test your model's goodness-of-fit? (d) Did you test other assumptions of logistic regression and assess potential multicollinearity issues? You may want to discuss these items in Section 2.3 to ensure reproducibility and transparency of your research.
3. Results, Lines 185-215. I felt your text here was repetitive of what was already listed in Table 1. Could you cut the text down to only talk about the most important points? This could help the reader effortlessly navigate the paper and avoid redundant information.
4. Tables 1-4. In your table captions, could include the sample size for each? This is good practice because we want tables to be standalone and easily interpreted without having to read the rest of the paper.
5. Discussion, Lines 313-317. I think you landed on some important points here, but it could use more detail and nuance here. The prevalence of walking and light outdoor exercise also increased because it was a way to deal with boredom and other unwanted psychological symptoms. This could be mentioned and also paired with a recommendation that governments should be careful about imposing restrictions (e.g., closing recreational facilities and playgrounds) that prevent individuals from engaging in desirable health behaviours (i.e., physical activity).
6. Discussion, Line 332. Could you add a subheading here called "4.1. Limitations" It would be nice to separate so the reader can immediately jump to it if needed.
7. Limitations. What about other regions? Since you only investigate data from Gyeongnam province, you should state that findings could be different in other provinces.
8. Lines 341-346. Sorry, I didn't quite understand what you were trying to communicate here. Could you revise this paragraph?
7. Abstract and Conclusion. Your conclusions are not wrong, but they are too broad and don't add much to the literature. Active interventions, appropriate management, and clear guidelines are important, but you should be more specific regarding your data in terms of what you wish recommendations for future infectious disease outbreaks should be. For instance, you noted that alcohol consumption, depressive symptoms and physical activity were affected during the pandemic. What would you suggest governments do? There are so many examples such as: increased mental health and alcohol use supports (e.g., digital resources) alongside restrictions, consider physical health implications during policy implementation, making exceptions to restrictions to continue promoting physical activity during disease outbreaks, etc. You can write whatever you wish, but please be specific and revise your abstract and conclusion section.
I think you paper could benefit from extensive copyediting and multiple readings because it contained several errors and some overly simplistic language at times. Please see below for some examples:
1. Lines 97-98. Add an "=" symbol, by changing to "male = 44.2%, female = 55.8%) and "(male = 44.9%, female 55.1%)".
2. Tables 2 and 4. Your column headings and row names are in lower case. Please change the headings to capitalize the first letter (e.g., "Variable, "Odds ratio", "Smoking"). You also have a second column which is unnamed, instead merge your "Variable" column with the unnamed one so it reads like this:
Smoking
Current smoking proportion
Current smoking proportion (male)
Smoking cessation proportion
3. Lines 320-321. I think you accidentally started a new paragraph here? "...after COVID-19 onset. [new paragraph] This could be due to..."
4. Line 369. You're missing a period at the end of the final sentence of your conclusion section.
5. Line 46. Change "was" to "The elderly were particularly.."
6. Line 104. Change "2.2.1 Dependente variables" to "2.2.1 Dependent variables"
Author Response
a

Round 2
Reviewer 1 Report
The manuscript was improved and could be published
Reviewer 3 Report
Thank you for revising the manuscript. I believe it is much stronger than before.